# Production of Siamenoside Ⅰ and Mogroside Ⅳ from *Siraitia grosvenorii* Using Immobilized β-Glucosidase

**DOI:** 10.3390/molecules27196352

**Published:** 2022-09-26

**Authors:** Hung-Yueh Chen, Ching-Hsiang Lin, Chih-Yao Hou, Hui-Wen Lin, Chang-Wei Hsieh, Kuan-Chen Cheng

**Affiliations:** 1Institute of Food Science and Technology, College of Bioresources and Agriculture, National Taiwan University, Taipei 10617, Taiwan; 2Department of Seafood Science, National Kaohsiung University of Science and Technology, Kaohsiung 81157, Taiwan; 3Department of Optometry, Asia University, 500, Lioufeng Rd., Wufeng, Taichung 41354, Taiwan; 4Department of Food Science and Biotechnology, National Chung Hsing University, Taichung 40227, Taiwan; 5Institute of Biotechnology, College of Bioresources and Agriculture, National Taiwan University, Taipei 10617, Taiwan; 6Department of Medical Research, China Medical University Hospital, China Medical University, 91, Hsueh-Shih Road, Taichung 40402, Taiwan

**Keywords:** *Siraitia grosvenorii*, saponins, β-glucosidase, enzyme immobilization, glutaraldehyde

## Abstract

*Siraitia grosvenorii* is a type of fruit used in traditional Chinese medicine. Previous studies have shown that the conversion of saponins was often carried out by chemical hydrolysis, which can be problematic because of the environmental hazards it may cause and the low yield it produces. Therefore, the purpose of this study is to establish a continuous bioreactor with immobilized enzymes to produce siamenoside I and mogroside IV. The results show that the immobilization process of β-glucosidase exhibited the best relative activity with a glutaraldehyde (GA) concentration of 1.5%, carrier activation time of 1 h and binding enzyme time of 12 h. After the immobilization through GA linkage, the highest relative activity of β-glucosidase was recorded through the reaction with the substrate at 60 °C and pH 5. Subsequently, the glass microspheres with immobilized β-glucosidase were filled into the reactor to maintain the optimal active environment, and the aqueous solution of *Siraitia grosvenorii* extract was introduced by controlling the flow rate. The highest concentration of siamenoside I and mogroside IV were obtained at a flow rate of 0.3 and 0.2 mL/min, respectively. By developing this immobilized enzyme system, siamenoside I and mogroside IV can be prepared in large quantities for industrial applications.

## 1. Introduction

*Siraitia grosvenorii* has been used as a traditional Chinese medicinal ingredient to treat lung fever and sore throat, etc. In addition to basic nutrients, including carbohydrates, proteins, amino acids and vitamins, *Siraitia grosvenorii* contains lots of flavonoids, phenols and terpenoids. These compounds represent the biochemical basis of *Siraitia grosvenorii*, which possesses various medicinal properties [1]. *Siraitia grosvenorii* contains a proportion of saponins with cucurbitane-type triterpenoids called mogrosides that are composed of several glycosylated saccharides linked to the non-glycosyl moiety with β-bonds [2]. Mogrosides have a number of biological activities, such as antioxidant, anti-inflammatory, anti-cancer and anti-infection properties, etc., and have great potential to be developed into biopharmaceuticals [1]. Among them, mogroside V is the main type making up the greatest proportion of content, accounting for about 0.8–1.3% (*w*/*w*) [3]. Compared with the other mogrosides, siamenoside I has a slightly lower physiological activity, but is the sweetest among all mogrosides [4,5]. Mogroside IV and siamenoside I are two intermediate products upon mogroside V hydrolyzation, which is further hydrolyzed into mogroside IIIE by β-glucosidase [6].

β-D-Glucosidase (E.C. 3.2.1.21) mainly catalyzes β-D-glucopyranosides from β-1,4 and β-1,6 of non-reducing sugar ends [7]. β-D-Glucosidase is widely used in the food, medicine and biomass energy industries. The main principle is to use the process of hydrolysis to produce glucose and the required substances, such as those that engender better aroma in grape juice and red wine [8], or the daidzin and glycitin in soy milk are cleaved from the glycosidic bond to generate non-glycosidic daidzein and glycitein [9]. The immobilized enzyme has the following characteristics: (1) the resistance to temperature and pH value is improved; (2) the storage stability is enhanced; (3) the enzyme can be reused in different reaction environments; and (4) the enzyme activity can be improved by reacting in an organic solvent [10].

Since mogroside IV, mogroside V and siamenoside I have different physiological activities, in this study, in addition to optimizing the β-D-glucosidase immobilized system through an adjustment of immobilizing conditions, a continued bioreactor was designed to control the production of various mogrosides by simply adjusting the flow rate. The continued bioreactor is a type of packed bed reactor, which has been extensively applied in food manufacturing and processing for different purposes, such as hydrolyzing lactose in milk to produce lactose-free milk [11], hydrolyzing sucrose to produce syrup with higher fructose content [12], and clarifying apple juice [13].

## 2. Results and Discussion

### 2.1. Morphology of Glass Microsphere

The β-glucosidase was immobilized on glass microspheres using glutaraldehyde (GA) as the cross-linker. To determine the situation of the immobilization, a scanning electronic microscope (SEM) was used to observe the surfaces of glass microspheres with and without β-glucosidase immobilization (Figure 1A–C). The interaction between carriers and the enzyme provides specific chemical, physical, biochemical and kinetic properties for each immobilized enzyme [14].

In general, when the diameter of the carrier is smaller, the specific surface area will increase, resulting in more enzymes being immobilized. A carrier with a smaller size will reduce the diffusion limitation, which usually decreases the reduction of enzyme activity [15]. The surfaces of glass microspheres without β-glucosidase were smooth and the average diameter was about 8 μm. After β-glucosidase immobilization through covalent binding with GA, the surfaces of glass microspheres became rough and granulated with a lot of flocci attached to them, as indicated by the red arrow. The morphology results showed that β-glucosidase was successfully immobilized on the glass microspheres through GA modification.

### 2.2. Verification of Enzyme Immobilisation

Figure 1D,E shows the ESCA spectra of glass microspheres with and without β-glucosidase immobilization, which were used to investigate the chemical bonding between the enzyme and the carriers. The glass microspheres were cross-linked with the β-glucosidase using GA as the cross-linker. GA is a type of cross-linking agent that is commonly used due to its high commercial availability, colorlessness, low cost, water solubility and, most importantly, quick reaction with an amine group on an enzyme [16]. Before immobilization, the characteristic signals at 284.6, 285.8, 286.2 and 288.9 eV corresponded to C-C, C=N, C-OH and C=O, respectively. After immobilization, the signal at 286.2 eV was reduced, and the signal at 285.8 eV was enhanced, which means that β-glucosidase was successfully linked on the surfaces of glass microspheres by GA and formed the Schiff base [17]. According to the results, the β-glucosidase was successfully immobilized on the glass microspheres using GA as the cross-linker.

### 2.3. Determination of Optimal Conditions for the β-Glucosidase Immobilized System

Using GA as the crosslinker significantly affected the enzyme activity, which includes the GA concentration, the activation time and the coupling time [15]. Figure 2A–C shows that β-glucosidase had the maximal relative activity, in which the GA concentration was 1.5%, the activation time was 1 h and the coupling time was 12 h. More GA concentration, activation time and coupling time may change the enzyme structure due to excessive crosslinking between GA and enzyme, which in turn decreases the enzyme activity [18].

To optimize the catalytic activity, the reaction temperature and the pH value were considered, both of which significantly affected the enzyme activity. In order to determine the optimal reaction conditions for the immobilized system, the catalytic activity of β-glucosidase was evaluated at different reaction temperatures and pH values. The results for the reaction temperature show that the relative activity of β-glucosidase with and without immobilization was the same at 60 °C (Figure 2D). However, when the reaction temperature changed, the relative activity of the free enzyme changed more drastically than the immobilized enzyme, which means that the resistance to the reaction temperature was increased after enzyme immobilization. That may be because the immobilized enzyme reduced the flexibility and thermal vibration of the configuration, and then reduced the probability of protein denaturation [19]. Just as the relative activity was recorded at its highest at the same reaction temperature of 60 °C both with and without immobilization, β-glucosidase also demonstrated the greatest relative activity at the same pH value of 5 under the two conditions (Figure 2E). Based on the results, a reaction temperature of 60 °C and a pH value at 5 were be used for subsequent experiments.

### 2.4. Determination of Kinetic Parameters for β-Glucosidase Immobilized System

The kinetic parameters, Vmax and Km represent the affinity between the enzyme and the substrate, which was calculated by the reaction time and the catalytic activity. These Michaelis–Menten kinetic constants are important characteristics to compare different enzyme systems [20]. As shown in Figure 3 and Table 1, the Vmax and Km of free β-glucosidase were 5.15 mM/min and 2.36 mM, respectively, and the Vmax and Km of immobilized β-glucosidase were 1.04 mM/min and 3.31 mM, respectively. After enzyme immobilization, the Vmax was decreased, and the Km was increased due to changes in the enzyme structure and the diffusion limitation [21]. Table 1 also shows the kinetic parameters of τ_50_ and τ_complete_ with and without immobilization, in which τ_50_ represents the time required for half of the reaction to complete and τ_complete_ represents the time required for the reaction to complete. Both parameters are calculated by plotting the relative activity at different reaction times. As shown in the results, the τ_50_ of the free enzyme was 1.33 min and the τ_complete_ was 4.43 min, whereas the τ_50_ of the immobilized enzyme is 6.43 min and the τ_complete_ is 21.36 min. The results of τ_50_ and τ_complete_ were similar to those for Vmax and Km, the catalytic time of immobilized β-glucosidase was increased. That was due to the diffusion limitation of immobilized β-glucosidase and the reduced probability of the substrate binding to the enzyme active site [11].

### 2.5. Storage Stability and Reusability of the β-Glucosidase Immobilized System

Enzyme immobilization has attracted much attention in recent years, due to the enzyme storage stability and reusability reducing costs and increasing environmental sustainability [22]. Figure 4A shows the results of the reusability of immobilized β-glucosidase by covalent bonding to glass microspheres as the carriers. The immobilized β-glucosidase still showed 80% relative activity after four catalysis reactions, and retained 50%.

The parameters τ50 and τ100 represent the time needed to catalyze 50% and 100% of the p-NPG, respectively.

Relative activity after 10 times of catalysis. Previous studies reported that the relative activity of β-glucosidase immobilized on chitosan was lower than 50% by reusing the reaction seven times [23]. In contrast, β-glucosidase immobilized on glass microspheres effectively enhanced the enzyme activity. The catalytic activity of β-glucosidase was decreased in the subsequent cycles, due to the conformation changing of enzymes, the blocking of some reaction substrate, the loss of carriers and the removing of enzyme from carriers [24]. The storage stability of immobilized β-glucosidase is shown in Figure 4B, the relative activity of immobilized β-glucosidase was above 90% after storage at 4 °C for 35 days, which indicates that this immobilization system maintained the superior stability and allowed long-term operation. These results suggest that β-glucosidase immobilized on the glass microspheres had increased storage stability and reusability, bearing out excellent potential for industrial applications.

### 2.6. Production of Siamenoside I and Mogroside IV from Siraitia grosvenorii through the β-Glucosidase Immobilized Bioreactor

The β-glucosidase immobilized bioreactor was adopted according to the design diagram of the reactor (Figure 5A) and filled with glass microspheres, and the actual diagram is shown in Figure 5B. There are two types of bioreactor systems, namely suspension and immobilization. Some reactors use a combination of these two systems to achieve the best culture mode by taking into account the advantages of both, for example, immobilization of cells or enzymes on a carrier in a stirring, gas-lifting or bubbling reactors [25]. Through comprehensive comparison of all types, the immobilization system known as a packed bed type is considered to be the most suitable for applications relating to enzyme reaction due to its design simplicity and low cost. The methodology used in a packed bed type involves fixing enzyme to carriers in the reactor’s column and allowing the reaction solution to flow from the bottom of the column to the top.

The operation efficiency of a packed bed type reactor is high since the carrier is not subjected to mechanical shear force. Moreover, it is commonly used in enzyme-catalyzed reactions that consist of different phases of solid and liquid [26]. The results of each flow rate to convert *Siraitia grosvenorii* extract into mogroside IV, mogroside V, mogroside IIIE, and siamenoside I are shown in Figure 6. The total amount of *Siraitia grosvenorii* extract at different flow rates had no significant enhancement, which means that the alteration and the catalysis of different *Siraitia grosvenorii* extracts had no inference from variables of this bioreactor design. For the mogroside V production, the amount of mogroside V was enhanced according to the increase in the flow rate. Given that mogroside V was the precursor of other mogrosides, the faster the flow rate (meaning the shorter the reaction time), the more mogroside V was produced. On the other hand, the production of mogroside IIIE as the final product was the opposite of mogroside V. The slower the flow rate (meaning the longer reaction time), the more mogroside IIIE was generated. In addition, the two intermediate products, siamenoside I and mogroside IV, had the highest production at flow rates of 0.3 mL/min and 0.2 mL/min, respectively. In conclusion, the production of these mogrosides, especially siamenoside I and mogroside IV, was significantly influenced by the flow rate, and the bioreactor designed in this experiment could control the flow rate to obtain two intermediates with higher conversion rates.

## 3. Materials and Methods

### 3.1. Materials

*Siraitia grosvenorii* was purchased from Huang Changsheng Traditional Chinese Medicine Store in Taipei City. p-Nitrophenylglucopyranose (p-NPG), β-glucosidase, glass particles, sodium carbonate, disodium hydrogen phosphate, sodium dihydrogen phosphate, glutaraldehyde (GA), nitric acid, 3-aminopropyltriethoxysilane (3-APES), formic acid, Bio-rad reagent, and hydrocorticoid were purchased from Sigma-Aldrich Co. (St. Louis, MO, USA). Methanol and ethanol were purchased from ECHO Chemical Co., Ltd. (Miaoli, Taiwan).

### 3.2. Enzyme Immobilization

To prepare carriers, glass microspheres were prepared following a modified version of the method by Chen et al.; a total weight of 2 g glass microspheres was first treated with 40 mL 10% nitric acid (HNO_3_) at 90 °C for 1 h, and then washed several times with distilled water. After the first treatment, glass microspheres were treated with 40 mL 10% 3-amino-propyltriethoxysilane (APES) aqueous solution at 70 °C for 3 h, then washed several times with distilled water and stored at 4 °C [23,27]. For the activation of glass microspheres, modified version of the method by Chen et al. was used, a total weight of 2 g glass microspheres was treated with 1.5% (*w*/*v*) GA and stirred at 100 rpm at room temperature for 1 hr, then washed three times with distilled water. After activation, glass microspheres were treated with 1% (*w*/*v*) β-glucosidase solution at 4 °C for 12 h, then washed three times with 0.1 M phosphate buffer and stored at 4 °C. The concentration of immobilized β-glucosidase was calculated by the Bradford method [28].

### 3.3. Morphology Characterization

In this study, p-NPG was used as the substrate to evaluate the catalytic efficiency of β-glucosidase in various systems (with and without immobilization). The glass microspheres with and without β-glucosidase immobilization were dried by lyophilizer, then the surface morphologies were evaluated using scanning electron microscopy (SEM, Model JSM-6300, JEOL, Tokyo, Japan) at an accelerating voltage of 10 kV after gold sputtering. The covalent bonding between glass microspheres, GA and β-glucosidase was confirmed by electron spectroscopy for chemical analysis (ESCA, VG MICROTECH, MT-500, British).

### 3.4. Determination of Reaction Conditions

The determination of the optimal reaction temperature was evaluated by 0.5 mg/mL p-NPG as a substrate in pH 4 citric acid buffer at 30, 40, 50, 60, 70 and 80 °C for 10 min. The optimal reaction pH value was examined by 0.5 p-NPG as a substrate at 60 °C in different pH values (from 4 to 8). The β-glucosidase activity was calculated by measuring the absorbance at 425 nm using a Multiskan GO microplate spectrometer (Thermo Fisher Scientific, Waltham, MA, USA) [29].

### 3.5. Determination of Kinetic Parameters

To determine kinetic parameters of β-glucosidase, including *K*, τ_50_, τ_complete_, the Michaels–Menten kinetic constant (Km) and maximal velocity (Vmax), different concentrations of p-NPG (from 0 to 20 mM) as the substrate for the reaction of free and immobilized β-glucosidase for different reaction times (from 0 to 90 min) were used. The parameters of *K*, τ_50_ and τ_complete_ represents the rate constant (min^−1^), the time required to reach required p-NPG and the time necessary to complete p-NPG, respectively. The β-glucosidase activity time curve fitted Equation (1), and the Vmax and Km were calculated based on the Lineweaver–Burk plot (Equation (2))
Abs(t) = Abs(S)(1 − e^−Kt^)(1)
V = Vmax [S]/Km + [S](2)

### 3.6. Storage Stability and Reusability

The evaluation of the storage stability was determined for the immobilized β-glucosidase activity for 0, 2, 4, 6, 8, 10, 15, 20, 25, 30 and 35 days storage at 4 °C. The reusability was determined by running reactions using immobilized β-glucosidase 10 times in 1 day. Both reusability and storage stability were reacted under the optimal conditions, and β-glucosidase activity in the first reaction and day 0 was a relative activity of 100%.

### 3.7. Creation of the β-glucosidase Immobilized Bioreactor

As in the β-glucosidase immobilized bioreactor shown in Figure 5, the column was fixed by the rack. The outer layer of the column was connected to a constant temperature water bath to maintain the reaction temperature, and the inner layer was full the immobilized glass microspheres. For the β-glucosidase catalysis, three times volume of *Siraitia grosvenorii* extract was passed into the reactor from below and collected from above through the pump, the final product was collected in a bottle at 4 °C to terminate the enzyme activity.

### 3.8. Siraitia Grosvenorii Extraction

The extraction of *Siraitia grosvenorii* was performed using a modified version of the method by Zhang et al., *Siraitia grosvenorii* was broken by a grinder, screened by 60 mesh, and then stored at room temperature in the drying oven. To extract *Siraitia grosvenorii*, 15 g *Siraitia grosvenorii* powder was added to pH 5 citric acid buffer at 100 °C for 10 min, then filtered by suction and the supernatant stored as the *Siraitia grosvenorii* solution for subsequent use. For the bioreactor system, 27 mL *Siraitia grosvenorii* solution was added into the column at different flow rates (from 0.1 to 035 mL/min) [30].

### 3.9. HPLC Analysis

The analytic process was modified according to Shen et al., the resolution of *Siraitia grosvenorii* extract, which included siamenoside I and mogroside IV, was performed by a YMC-Pack-ODS-AMC C18 column (5 mm, 250 × 4.6 mm) attached to a high performance liquid chromatography (HPLC) system containing the pump (PU-2089, JASCO, Tokyo, Japan) and the analytical mixer. The results were analyzed using a SISC chromatography data system (SISC, New Taipei City, Taiwan). The detection was performed by UV absorption at 210 nm, and the injection volume and the elation rate were 20 mL and 0.6 mL/min, respectively. A 0.01% formic acid solution and methanol were used as the eluent [31].

### 3.10. Statistical Analysis

All experiments are expressed as mean ± standard deviation and performed at least in triplicate. The data analysis was measured using Prism, Minitab, Excel and Sigma plot, and the statistical analysis was performed by ANOVA and Fisher’s LSD. The *p*-value was set at 0.05.

## 4. Conclusions

The optimal process for β-glucosidase immobilization was successfully developed with 1.5% GA concentration for 1 h activation, and coupled with β-glucosidase for 12 h. According to the results as presented, although the optimized conditions of both reaction temperature and pH value were not significantly affected by whether enzyme immobilization was carried out, the resistance of β-glucosidase to temperature was increased after immobilization. Moreover, the kinetic parameters of immobilized β-glucosidase were worse than free enzyme, due to the diffusion limitation and the conformation change after GA covalent bonding. Because of the advanced resistance to reaction conditions, the gainful storage stability of 35 days and the usability of more than 10 times were recorded. The increasing amount of research into mogrosides has pointed out that siamenoside I is 563 times sweeter than 5% sucrose aqueous solution and has a better flavor than other types of mogrosides, making it more suitable for use as a sweetener. Also, mogroside IV can slow down the symptoms of pulmonary fibrosis and inhibit the proliferation of cancer cells, thus has great potential in the biopharmaceutical industry. In this study, the β-glucosidase immobilization was optimized by GA concentration, activation time, coupling time, reaction temperature and pH value. Moreover, this study establishes a β-glucosidase immobilization system for the continued reaction of different mogrosides by controlling the flow rate. Therefore, this immobilized β-glucosidase system has the potential to be applied in industry.

## Figures and Tables

**Figure 1 molecules-27-06352-f001:**
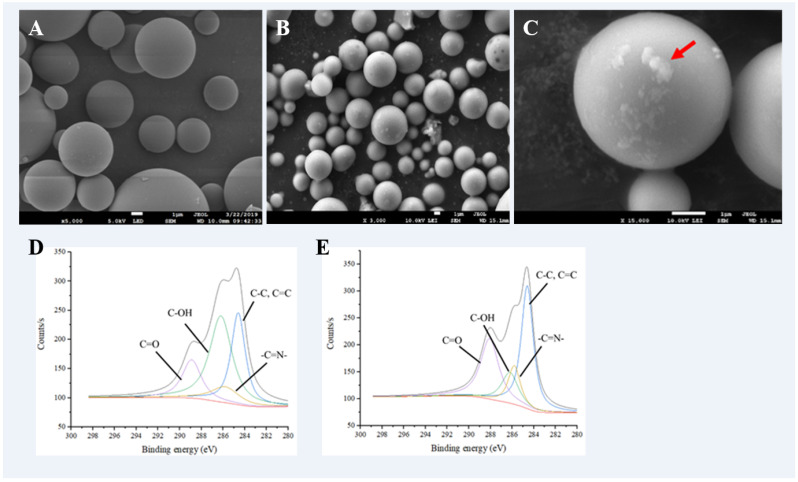
SEM images and the C element scan of glass microspheres without and with β-glucosidase immobilization. The SEM images of glass microspheres without β-glucosidase immobilization at 5000× magnification power (**A**), and with β-glucosidase immobilization at 3000× (**B**) and 15,000× (**C**) magnification power. The C element scan without (**D**) and with (**E**) β-glucosidase immobilization by ESCA. The characteristic signals at 284.6, 285.8, 286.2 and 288.9 eV corresponded to C-C, C=N, C-OH and C=O, respectively. The red arrow is pointing the attached β-glucosidase.

**Figure 2 molecules-27-06352-f002:**
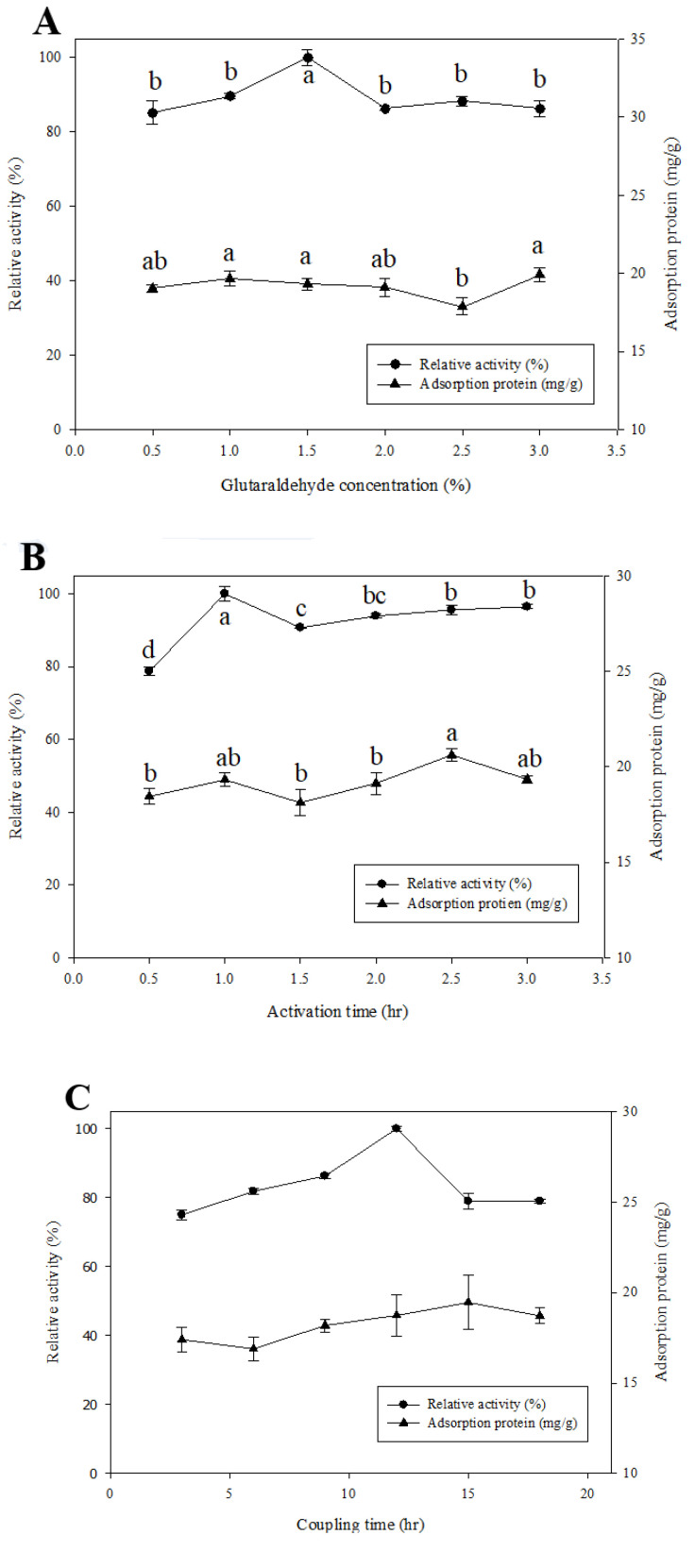
Immobilization conditions of β-glucosidase onto glass microspheres. The relative activity with glutaraldehyde concentration (**A**), the relative activity for activation time (**B**), and the relative activity for coupling time (**C**). The optimal reaction temperature (**D**) and pH value (**E**) on the relative activity of free and immobilized β-glucosidase. Each value is expressed as mean ± SD, and values marked by different letters were significantly different by LSD tests (*p* < 0.05).

**Figure 3 molecules-27-06352-f003:**
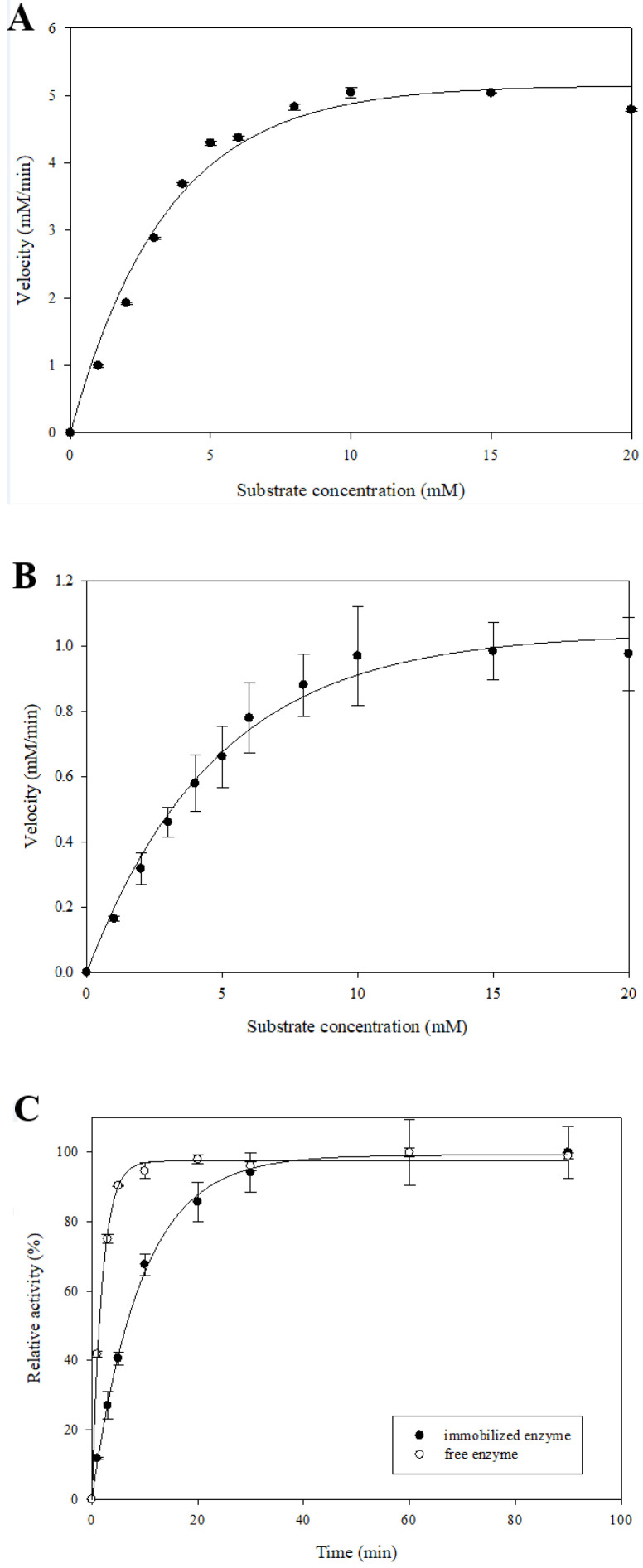
Effects of p-NPG concentration on the activity of free β-glucosidase (**A**) and immobilized β-glucosidase (**B**), and effects of time on the activity of free enzyme and immobilized β-glucosidase (**C**). Each value is expressed as mean ± SD, and values marked by different letters are significantly different by LSD tests (*p* < 0.05).

**Figure 4 molecules-27-06352-f004:**
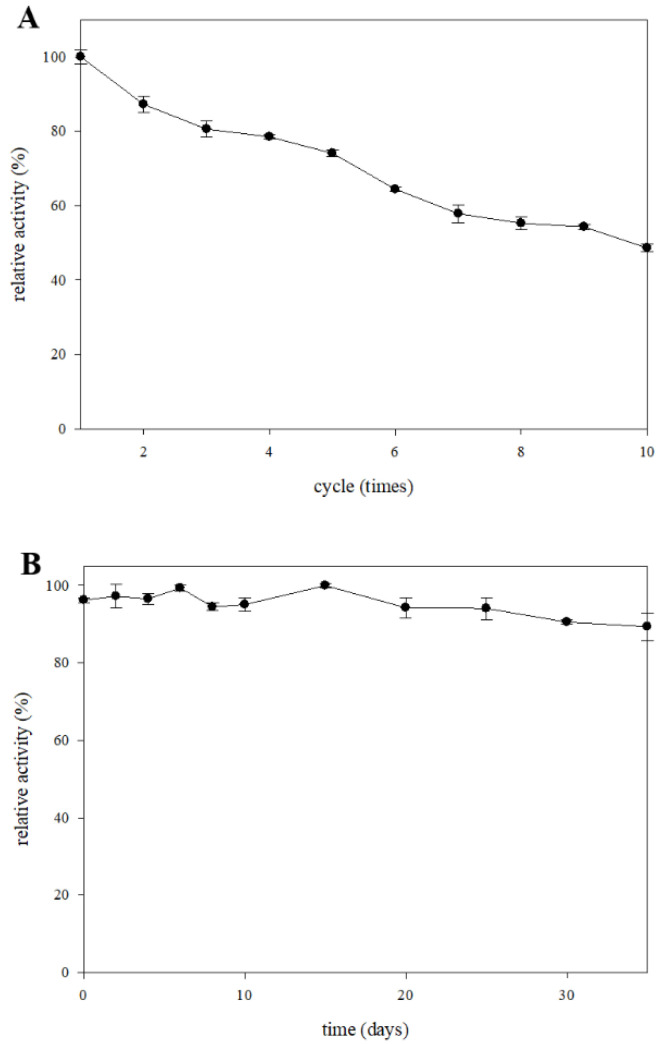
The usability (**A**) and the storage stability (**B**) of immobilized β-glucosidase. Each value is expressed as mean ± SD.

**Figure 5 molecules-27-06352-f005:**
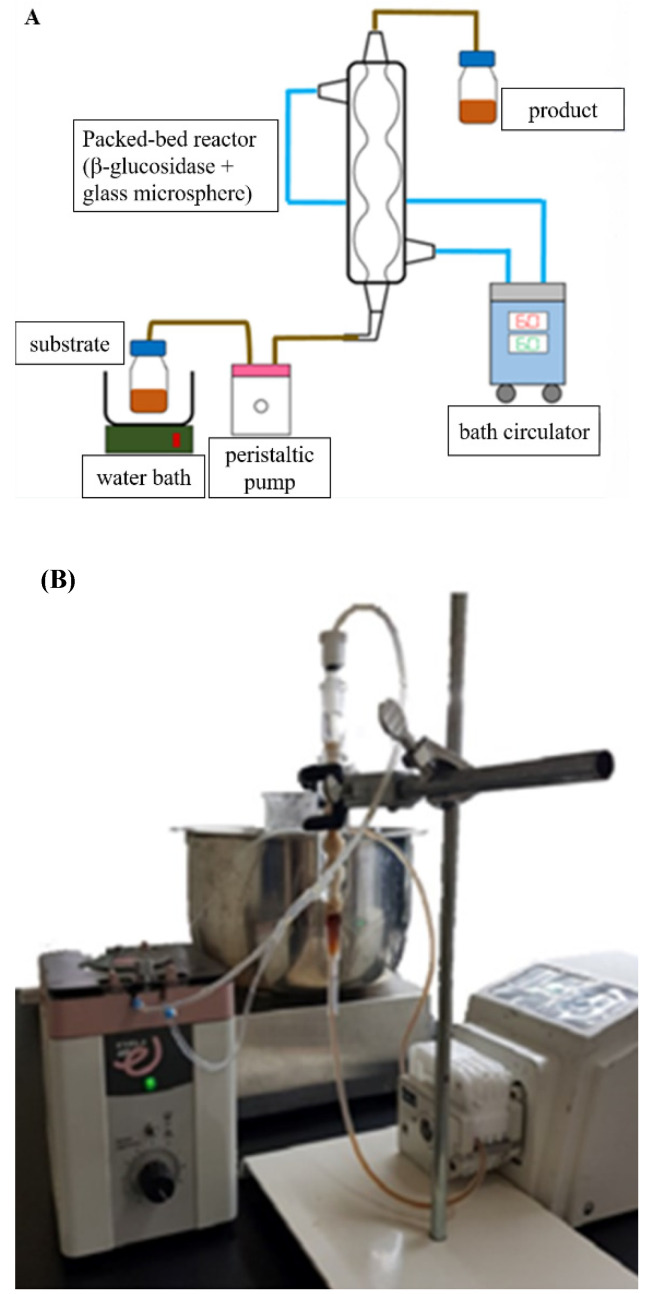
The design of (**A**) and the real (**B**) immobilized β-glucosidase bioreactor. The inner layer of the column (9 mL) with 2 g β-glucosidase immobilized glass microspheres and the outer layer full, with constant temperature water to control the reaction temperature. The sample injected into the column from the bottom through a pump and collected by a bottle with a constant temperature water tank for the termination of the catalysis.

**Figure 6 molecules-27-06352-f006:**
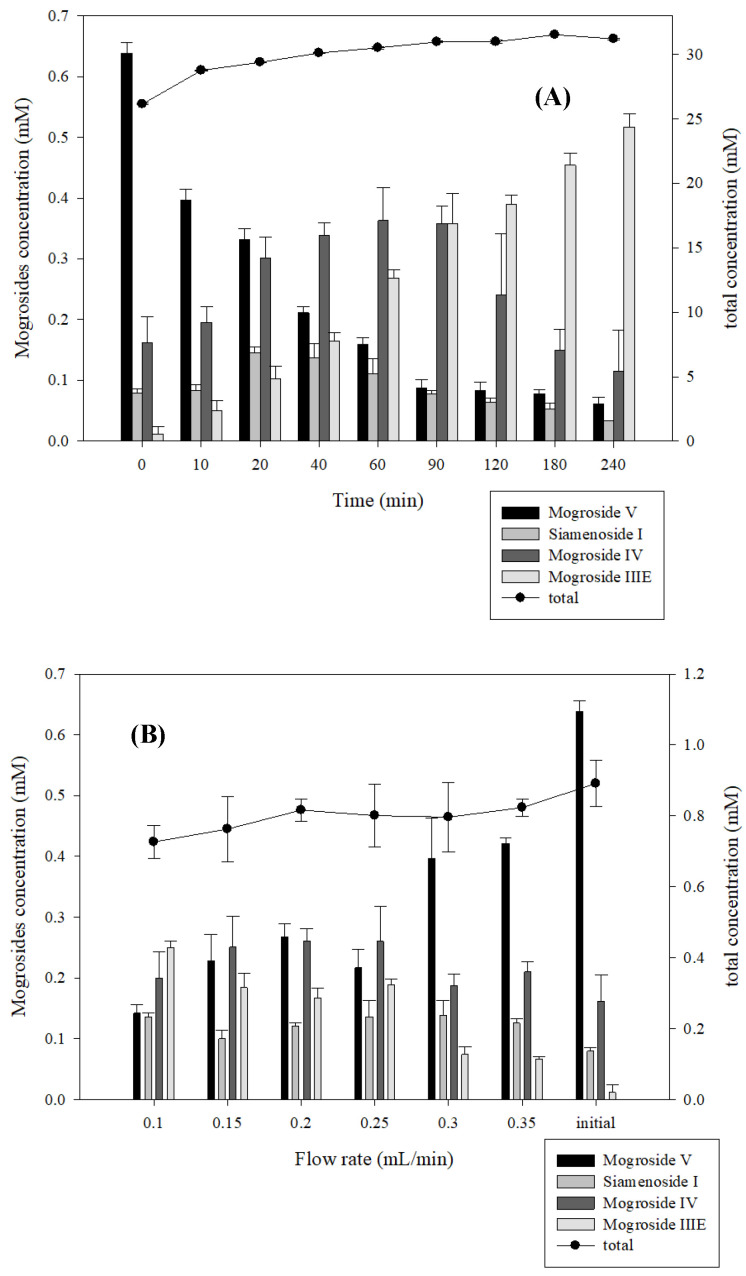
The result of immobilized β-glucosidase converting mogrosides at different times (**A**), and using the bioreactor to convert mogrosides by different flow rates (**B**). Each value is expressed as mean ± SD.

**Table 1 molecules-27-06352-t001:** Enzyme kinetic parameters of free and immobilized β-glucosidase.

	*K* (mM/min)	τ_50_ (min)	τ_complete_ (min)
Free β-glucosidase	0.52	1.33	4.43
Immobilized β-glucosidase	0.11	6.43	21.36

## Data Availability

The data presented in this study are available on request from the corresponding author.

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
