# Peer review of "Production of Siamenoside I and Mogroside IV from Siraitia grosvenorii Using Immobilized β-Glucosidase"

_molecules, 2022, doi:10.3390/molecules27196352_

Round 1

Reviewer 1 Report

·       All minor remarks are highlighted in the manuscript.

Author Response

1. All minor remarks are highlighted in the manuscript. Thank you very much for your valuable comments. We have revised our manuscript in Word file and it has been uploaded for your perusal.

Reviewer 2 Report

The authors should change the keywords because they are repeated in the title. 

I recommend that the authors should reinforce the statement of novelty and originality. 

The authors should check so well state of the art on this topic because there are few references in the intro section (9 references). 

The main goal is unclear. 

The authors should improve the quality of the figures. 

I recommend that the authors reinforce the discussion section because it is poor.

Author Response

  1. The authors should change the keywords because they are repeated in the title. 

Thank you very much for your valuable comments. We removed some words and modified our keywords. The modified keywords are Siraitia grosvenorii; saponins; β-glucosidase; enzyme immobilization; glutaraldehyde.

  1. I recommend that the authors should reinforce the statement of novelty and originality. 

We rewrote Conclusions (line 356-360) to clarify the novelty and originality as follows:

In this study, the β-glucosidase immobilization was optimized by GA concentration, activation time, coupling time, reaction temperature and pH value. Moreover, this study establishes a β-glucosidase immobilization system for the continued reaction of different mogrosides production by controlling the flow rate.

  1. The authors should check so well state of the art on this topic because there are few references in the intro section (9 references).

We added the references as follows:

[10] Chen, H.-Y.; Ting, Y.; Kuo, H.-C.; Hsieh, C.-W.; Hsu, H.-Y.; Wu, C.-N.; Cheng, K.-C., Enzymatic degradation of ginkgolic acids by laccase immobilized on core/shell Fe3O4/nylon composite nanoparticles using novel coaxial electrospraying process. International Journal of Biological Macromolecules 2021, 172, 270-280.

[11] Ko, C.-Y.; Liu, J.-M.; Chen, K.-I.; Hsieh, C.-W.; Chu, Y.-L.; Cheng, K.-C., Lactose-free milk preparation by immobilized lactase in glass microsphere bed reactor. Food Biophysics 2018, 13, (4), 353-361.

[12] Amaya-Delgado, L.; Hidalgo-Lara, M.; Montes-Horcasitas, M., Hydrolysis of sucrose by invertase immobilized on nylon-6 microbeads. Food chemistry 2006, 99, (2), 299-304.

[13] de Oliveira, R. L.; Dias, J. L.; da Silva, O. S.; Porto, T. S., Immobilization of pectinase from Aspergillus aculeatus in alginate beads and clarification of apple and umbu juices in a packed bed reactor. Food and bioproducts processing 2018, 109, 9-18.

[14] Homaei, A., Enzyme immobilization and its application in the food industry. Advances in food biotechnology 2015, 9, 145-164.

[15] Migneault, I.; Dartiguenave, C.; Bertrand, M. J.; Waldron, K. C., Glutaraldehyde: behavior in aqueous solution, reaction with proteins, and application to enzyme crosslinking. Biotechniques 2004, 37, (5), 790-802.

[26] Zhong, J. J., Recent advances in bioreactor engineering. Korean Journal of Chemical Engineering 2010, 27, (4), 1035-1041.

[27] Halim, S. F. A.; Kamaruddin, A. H.; Fernando, W., Continuous biosynthesis of biodiesel from waste cooking palm oil in a packed bed reactor: optimization using response surface methodology (RSM) and mass transfer studies. Bioresource technology 2009, 100, (2), 710-716.

  1. The main goal is unclear. 

We rewrote Introduction (line 63-70) to clarify the mail goal as follows:

In this study, in addition to optimizing the β-D-glucosidase immobilized system through an adjustment of immobilizing conditions, a continued bioreactor was designed to control the production of various mogrosides by simply adjusting the flow rate. The continued bioreactor is a type of packed bed reactor which has been extensively applied in food manufacturing and processing for different purposes, such as hydrolyzing lactose in milk to produce lactose-free milk [10], hydrolyzing sucrose to produce syrup with higher fructose content [11], and clarifying apple juice [12].

  1. The authors should improve the quality of the figures. 

We have resubmitted higher resolution figures.

  1. I recommend that the authors reinforce the discussion section because it is poor.

We have reinforced the discussion in Results and Discussions as follows:

Line 76-78: The interaction between the carrier and enzyme provides specific chemical, physical, biochemical and kinetic properties of each immobilized enzyme [14].

Line 116-118: GA is a type of cross-linking agent that is commonly used due to its high commercial availability, colorlessness, low cost, water solubility, and most importantly quick reaction with an amine group on enzymes [15].

Line 215-236: There are two types of bioreactor systems, namely suspension and immobilization systems. Some reactors use a combination of these two systems to achieve the best culture mode by taking into account the advantages of both, for example, immobilization of cells or enzymes on a carrier in a stirring, gas-lifting or bubbling reactors [26]. Through comprehensive comparison of all types, the immobilization system known as a packed bed type is considered to be the most suitable for applications relating to enzyme reaction due to its design simplicity and low cost. The methodology used in a packed bed type involves fixing enzyme to carriers in the reactor’s column, and allowing the reaction solution to flow from the bottom of the column to top. The operation efficiency of a packed bed type reactor is high since the carrier is not subjected to mechanical shear force. Moreover, it is commonly used in enzyme-catalyzed reactions that consist of different phases of solid and liquid [27].

Round 2

Reviewer 2 Report

I accept the manuscript to publish in Molecules.